# Chagas Disease: Detection of *Trypanosoma cruzi* by a New, High-Specific Real Time PCR

**DOI:** 10.3390/jcm9051517

**Published:** 2020-05-18

**Authors:** Simone Kann, Meik Kunz, Jessica Hansen, Jürgen Sievertsen, Jose J. Crespo, Aristides Loperena, Sandra Arriens, Thomas Dandekar

**Affiliations:** 1Department Research and Development, Bernhard-Nocht-Institute for Tropical Medicine (BNITM), 20359 Hamburg, Germany; hansen_jessica@gmx.de (J.H.); sievertsen@bnitm.de (J.S.); s.arriens@gmx.de (S.A.); 2Actually Medical Mission Institute, 97074 Wuerzburg, Germany; 3Chair of Medical Informatics, Friedrich-Alexander University of Erlangen-Nürnberg, 91054 Erlangen, Germany; meik.kunz@fau.de; 4Department Health Advocacy, Organization Wiwa Yugumaiun Bunkuanarrua Tayrona (OWYBT), Valledupar 200001, Colombia; josejosecrespog@gmail.com (J.J.C.); aristides.loperena@hotmail.com (A.L.); 5Department of Bioinformatics, Biocenter, Functional Genomics and Systems Biology Group, Julius-Maximilians University, 97070 Wuerzburg, Germany; dandekar@biozentrum.uni-wuerzburg.de

**Keywords:** Chagas disease, Chagas diagnosis, Chagas monitoring, Chagas real time PCR, *Trypanosoma cruzi*

## Abstract

Background: Chagas disease (CD) is a major burden in Latin America, expanding also to non-endemic countries. A gold standard to detect the CD causing pathogen *Trypanosoma cruzi* is currently not available. Existing real time polymerase chain reactions (RT-PCRs) lack sensitivity and/or specificity. We present a new, highly specific RT-PCR for the diagnosis and monitoring of CD. Material and Methods: We analyzed 352 serum samples from Indigenous people living in high endemic CD areas of Colombia using three leading RT-PCRs (k-DNA-, TCZ-, 18S rRNA-PCR), the newly developed one (NDO-PCR), a Rapid Test/enzyme-linked immuno sorbent assay (ELISA), and immunofluorescence. Eighty-seven PCR-products were verified by sequence analysis after plasmid vector preparation. Results: The NDO-PCR showed the highest sensitivity (92.3%), specificity (100%), and accuracy (94.3%) for *T. cruzi* detection in the 87 sequenced samples. Sensitivities and specificities of the kDNA-PCR were 89.2%/22.7%, 20.5%/100% for TCZ-PCR, and 1.5%/100% for the 18S rRNA-PCR. The kDNA-PCR revealed a 77.3% false positive rate, mostly due to cross-reactions with *T. rangeli* (NDO-PCR 0%). TCZ- and 18S rRNA-PCR showed a false negative rate of 79.5% and 98.5% (NDO-PCR 7.7%), respectively. Conclusions: The NDO-PCR demonstrated the highest specificity, sensitivity, and accuracy compared to leading PCRs. Together with serologic tests, it can be considered as a reliable tool for CD detection and can improve CD management significantly.

## 1. Introduction

Chagas disease (CD), also known as American Trypanosomiasis, is caused by the protozoan parasite *Trypanosoma cruzi* (*T. cruzi*). CD has been known for more than 100 years [1], but is still a challenging medical, economic, and social burden. Although prevention, control measures, and overall improvements in life quality have led to a decrease of incidence and prevalence of CD, it remains a major threat in Latin America. Above all, it is endemic in the central part and the southern half of the United States [2] and is expanding more and more to non-endemic countries such as Canada and Europe, mainly due to migration and travel [3,4,5,6,7,8,9]. According to the World Health Organization (WHO), about 6–7 million individuals are infected and 75 million live with the daily risk of infection [10]. Still, only less than 1% of the infected have access to diagnosis and treatment [11], leading to high mortality and morbidity rates. CD has two different phases. In the acute phase, the patient suffers from flu-like symptoms like fever, headache, and myalgias. Although the acute phase is the most promising one for a successful treatment [11,12], CD is mostly untreated [11] due to a lack of diagnostics and is declared as a fever of unknown origin. After the acute phase, the patient enters into the chronic phase, which either remains silent for years or even decades (indeterminate form), or leads to severe cardiac and/or gastrointestinal complications (cardiomyopathy, sudden heart death, megacolon, megaesophagus, etc.) in 30–40% of cases [6].

One key issue essential for the improvement of the current situation, is an effective tool to diagnose CD, in particular, a gold standard [13]. As this is not available, the WHO recommends two serologic tests with different techniques and to repeat or add a third test, if the results are discordant [14]. For congenital infections, a test after six months is suggested, when antibodies are produced by the child itself [14]. Although serologic tests serve well for the detection of chronic CD [15] (e.g., ELISA, immunofluorescence, rapid tests, etc.), they have their limitations. As they are mainly antibody based and antibody production needs several weeks, the patient has already entered the chronic phase when the serologic tests turn positive. This means that the acute cases cannot be detected by these tests, although this is of most importance, as early forms of CD infections have the best therapy outcomes [1,11,16,17,18,19]. Furthermore, as Chagas antibodies can remain positive for years, decades, or even a lifetime, and a new negative serologic test after infection and/or treatment is rare, it is difficult to assess the therapy outcome [16,17,18]. Reactivations and reinfections are also not assessable by using serology. Additionally, cross reactivity can interfere with result interpretations. Therefore, options to close these gaps are needed.

In many other diseases, real time polymerase chain reaction (RT-PCR) serves as a gold standard. Several PCRs have been developed for CD, but, as summarized in various published articles, problems occur in either sensitivity and/or specificity. Above all, they can vary in accuracy and performance and can have limitations in their reproducibility [3,15,19,20,21]. As PCRs are still important and give valuable information, the recommendation is to combine at least two PCRs to produce a reliable result [15,20,21]. Important reasons for the difficulties in developing a reliable PCR are the high genetic variability of *T. cruzi* (main types and subtypes, discrete typing units DTU’s I–VI), the size of the genome (>60 million base pairs, not all encoded in the database available), cross-reactions (e.g., with Leishmania spp. and the apathogenic *T. rangeli* [17,22]) as well as the technical factors (e.g., sample collection, extraction protocols, thermocycling) [20] and geographic specificities [15].

With the aim to improve CD diagnostics, we developed a novel Chagas RT-PCR (called newly developed one, NDO-PCR) and evaluated its capabilities using samples from high-endemic CD areas in Colombia and compared the results with leading PCRs. We wanted to fill the gap in serologic diagnostics and to overcome current PCR limitations to improve CD detection and management.

## 2. Material and Methods

### 2.1. Ethical Clearance

The study was approved by the Ethics Committee of Valledupar, César, Colombia. Informed consent was obtained from each participant or from the parent or legal guardian of a child before participation. The study was performed in accordance with the principles of the Declaration of Helsinki.

### 2.2. Study Design

The study focused on dengue fever and fever of unknown origin. The intention was to identify the underlying diseases. Particularly in regions of Colombia, the etiology and epidemiological characterization of these is difficult [23]. During the study, we also accessed regions of indigenous populations where fever-associated diseases are very common. Here, we saw that CD was one of the main problems. As each region has its own background interference factors, which might change specific results in tests, it is important to have a sero-negative group for comparison as well as to learn about the sero-prevalence of specific diseases. Consequently, in addition to patients with fever, afebrile, asymptomatic persons living in the same region and under the same conditions were included. With regard to the indigenous populations, all inhabitants of the villages were invited to participate. All participants answered a questionnaire (medical history, medication intake, vaccinations, actual complaints, etc.) and were physically examined before enrolment. Volunteers aged 12 years and above received a CD rapid test (RT) and a blood withdrawal if the RT was positive. The serum was then analyzed further with ELISA/immunofluorescence (IF) and the four different PCRs. From volunteers below 12 years of age, serum was taken directly, as acute infections of CD are usually acquired in childhood, and the RT/ELISA/IF and the four PCRs were performed independently of the serologic test outcome.

The amount of blood taken from the children was adjusted according to the guidelines for pediatric patients [24]. All participants received their results including a therapy recommendation, if appropriate.

### 2.3. Demographic Overview

The samples from the indigenous tribes (called Wiwas (334) and Wintukwas (18)) came from high endemic regions of César and La Guajira (Table 1). The average age was 20.17 years (from 351, one patient age unknown). A total of 182 samples were from female and 170 from male volunteers. The 158 samples were from children below 12 years, and 193 samples from volunteers 12 years and above.

### 2.4. Clinical Assessment

Medical anamnesis showed 98 (27.8%) patients stating complaints, with headache and diarrhea being the leading ones. The medical history was very sparse, as the indigenous participants could only very vaguely describe their reasons for hospitalization. A documentation of hospital stays could not be provided.

During the physical examination, 243 pathologic results were found. Leading complaints were cardiac, gastrointestinal, and orthopedic problems. Heart murmurs were found in 82 (23.3%) of the volunteers, and 24 (6.8%) patients complained about additional cardiac symptoms like palpitations, chest pain, etc. Other unspecific symptoms occurred in 25.3% (89 cases) such as dizziness, fatigue, etc. No findings were seen in 157 patients (44.6%).

### 2.5. Clinical Samples

In 2014, a retrospective study was performed using 352 serum samples collected from indigenous populations living in CD high-endemic regions of César (Tezhumake) and La Guajira (Ashintukwa, Marocaso, Siminke, Savannah Crespo, Arwamake, and Timake), in the northeast of Colombia. The samples were cooled in a specific cooling box at 4 °C (World Courier) and transported to the laboratory the same day, where extractions were made immediately. The sera and extracts were stored thereafter at −20 °C. International guidelines for airfreight sample transport were followed, permissions were gained from all involved parties, then a permanent cooling chain was controlled and confirmed. After arrival in Germany, the sera and extracts were stored at −80 °C until their further use for analysis.

## 3. Serologic Methods

All samples were tested by a Chagas Rapid Test (RT) (Chagas AB Rapid, Standard Diagnostics Inc. Bioline) and/or ELISA (Chagatest ELISA recombinant v. 4.0 Wiener Lab, Chagas IgG ELISA IBL), following the manufacturers’ protocols. Positive RTs and/or ELISAs were confirmed by indirect immunofluorescence (IF) on fixed epimastigotes of *T. cruzi* (Tulahuen strain TcII, from cell cultures) according to standard procedures [25].

### 3.1. DNA Extractions

Extractions were made from serum (200 µL) and finally eluted in 60 µL elution buffer following the instructions of the manufacturers’ protocols of the RTP Pathogen Kit (Stratec Molecular) and stored thereafter at −20 °C.

### 3.2. Real-Time-Polymerase Chain Reactions (RT-PCR) (kDNA, TCZ, and 18s rRNA)

The kDNA-RT-PCR targets the kinetoplast genome, also called minicircle and was chosen because it has been described as the best performing in previous studies [15] and most CD laboratories use it as an additional method. The TCZ-RT-PCR targets the mini-satellite TCZ region and 18S rRNA-RT-PCR targets the small subunit ribosomal RNA. They were chosen due to their results in specificity (100%) [15,20].

The kDNA TaqMan RT-PCR used the primers 32F, 5′-TTT GGG AGG GGC GTT CA-3′ and 148R 5′-ATA TTA CAC CCC CAA TCG AA-3′ and the TaqMan probe, 5′-CA TCTC AC CCG TACA TT-3′.

The 18S rRNA RT-PCR used the primers TcF1042, 5′-GCA CTC GTC GCC TTT GTG-3′ and TcF1144, 5′-AGT TGA GGG AAG GCA TGA CA-3′, and the TaqMan probe TCP1104, 5′-AA GAC CGA AGT CTG CCA ACA ACA C-3′.

kDNA- and 18S rRNA-PCRs were run with the cycling conditions reported by Qvarnstrom et al. (2012) [15].

The TCZ-PCR used the primers cruzi 1, 5′-AST CGG CTG ATC GTT TTC GA-3′ and cruzi 2, 5′-AAT TCC TCC AAG CAG CGG ATA-3′ and as a probe cruzi 3, 5′-CAC ACA CTG GAC ACC AA-3′, following the cycling conditions as described by Schijman et al. (2011) [20].

### 3.3. Sequencing

For sequence analysis, PCR amplicons (85 from kDNA-, one from TCZ- and one from NDO-PCR) were taken and used for plasmid vector preparation (TOPO^®^ TA Cloning^®^ Kit from Invitrogen, 76131 Karlsruhe, Germany). Five to 10 clones were picked and sequenced according to Big Dye Terminator version 3.1 (Life Technologies, BLAST query [26]. As the financial budget was limited, no additional sequencing could be performed.

### 3.4. Newly Developed One Real-Time Polymerase Chain Reaction (NDO-RT-PCR)

To find promising new sequences for primer and probe generation for the detection of *T. cruzi*, sequence analysis data of the 87 samples were aligned. At first, sequences positive for *T. cruzi* were compared to each other, then all sequences positive for *T. rangeli* were aligned to each other. Thereafter, a cross alignment was performed, which showed several eligible regions for a primer and probe construction. The most promising one was taken and showed a positive outcome after being tested. The new designed primer and probe set showed a high efficacy and efficiency and was compared to the leading ones (kDNA-, TCZ- and 18s rRNA- primer and probes).

The NDO-RT-PCR primers and probes including their cycler protocol were used as described in the patent publication from Kann et al. (2017) [27] and are summarized here in short: The NDO–RT-PCR used primer ChagasF 5′-GCACTATATTACACCAACCCC-3′ and ChagasR 5′-CATGCATCTCCCCCGTA-3′, and as a probe ChagasS 5′-CGAACCCCACCTCC-3′ (available TibMolBiol, Ref. No. 53-0755-96).

All primers and probes were purchased by Eurofins MWG, except for the probe for kDNA-PCR, which was obtained by Applied Biosystems. PCRs were run on a Rotor-Gene Q Cycler from Qiagen. Each sample was tested in triplicate. As a positive control, *T. cruzi* strain Tulahuen was used in all PCRs.

The three reference strains *Cl Brener*, *Y*, and *Brazil* were obtained from the strain collection of the National Reference Center for Tropical Agents at the Bernhard-Nocht Institute for Tropical Medicine (BNITM) in Hamburg. The trypanosomes were passaged once in BALB/c mice and were then frozen in aliquots. Aliquots were thawed and propagated in Nakamura medium as epimastigotes. The *Malaria* spp. and *Leishmania brasiliense* samples were from travelers consulting the BNITM, returning from Ghana and Brazil. *T. rangeli* (TrSA7) was taken, after being analyzed by sequencing, out of the gained sample collection of the study. All extractions were performed as described above.

### 3.5. Disease Classification

Acute or reactivated cases were defined by the direct detection of the parasite in the blood, thus a positive PCR run.

During the chronic phase, where only a few or no parasites are found in the blood, two different serological techniques are recommended by the WHO. Here, a Rapid Test and ELISA/IF was performed, as requested by the Colombians to entitle the patient to treatment. Chronic cases were divided into patients being positive in at least two serologic tests and negative in all PCRs and those being positive in at least two serologic tests and positive in at least one PCR run. The last group can be interpreted as acute, re-infected, re-activated, and/or (early) chronic phase of CD infection. As the NDO-PCR showed very high sensitivities in the detection rate, it can also detect low parasitaemias during the chronic phase.

### 3.6. Statistical Analysis

Data were warehoused using the phpMyAdmin software [28]. Statistical analysis was performed using the statistical program R. We calculated the following diagnostic parameters:
-Sensitivity: t_p/_t_p_ + f_n_-Specificity: t_n/_t_n_ + f_p_-Positive Predictive Value (PPV, precision): t_p/_t_p_ + f_p_-Negative Predictive Value (NPV): t_n/_t_n_ + f_n_-Accuracy (correct classification rate): t_p_ + t_n/_t_p_ + f_p_ + t_n_ + f_n_-False Negative Rate (FNR): f_n/_t_p_ + f_n_-False Positive Rate (FPR): f_p/_t_n_ + f_p_-Prevalence: t_p_ + f_n_/t_p_ + f_p_ + t_n_ + f_n_

(t_p_ = true positive, f_p_ = false positive, f_n_ = false negative, t_n_ = true negative).

Regression analysis was done in R, using the regression. Extrapolation provided the linear relationship assumed holds and was hence applied for the analysis. Here, the 87 sequenced PCR samples were taken as the gold standard and the PCR results of the other 265 samples were compared to those.

## 4. Results

### 4.1. Results of the NDO-PCR

The intra-assay showed a variation coefficient of 1.7% in 1:100 and of 2.4% in 1:1000 dilutions and the inter-assay showed a variation coefficient of 1.86%. The primer dimer run stayed negative. Primer and probes were tested with known strains such as CL. Brener (DTU TcII), Y (DTU TcII), and Brazil (DTU TcI) [29,30,31,32,33] and showed the expected positive outcomes. *Malaria* spp., *Leishmania Brasiliense*, and *T. rangeli* (TrSA7) was used for cross-reactivity testing, which did not occur. For detection limit determination, DNA from the positive control Tulahuen (DTU TcII) [34,35,36], taken from tissue culture, was determined in a serial dilution. The lowest dilution was defined as valid when 3/3 results were positive. The detection limit (LOD) of the NDO was compared to the kDNA-PCR one magnitude superior (1:100.000 versus 1:10.000; 1.5 copies/µL versus 15 copies/µL).

### 4.2. Sequenced Samples (87)

According to max score, total score, query cover, and expected value, the blasted sequences revealed 63 samples of *T. cruzi*, 14 of *T. rangeli*, and eight cases of *Homo sapiens* (unspecific human DNA, not Trypanosoma). The average identification correctness was 91.4%. In one sample, only the TCZ-PCR (and later also the NDO-PCR) showed a positive result. Additionally, this TCZ-PCR amplicon was sequenced as described above and revealed an infection with *T. cruzi*.

After the development of the NDO-PCR, another case occurred in which only the NDO-PCR showed a positive result. Sequencing of the NDO-PCR amplicon, performed as described above, presented *T. cruzi* as the underlying pathogen.

The NDO-PCR and the other leading PCRs were compared to this dataset, which was further used as the gold standard.

## 5. Serology

Out of the 87 sequenced samples, 72 were positive in one serologic test (RT/ELISA or IF), and 71 were positive in two tests (no sample was positive in two serologic tests and negative in all performed PCRs). Once the serology missed a positive chronic case, it was cross-reacted twice with *T. rangeli*, and seven times it reacted with unspecific DNA (*H. sapiens*). Only once did IF show a false positive result by marking *T. rangeli* as positive. In total, nine false positive and two false negative results occurred (verified by sequencing) (Table 2).

In the group of 265 non-sequenced samples, 112 were positive in at least two serologic tests. Overall, analysis by one or more serological test did not result in a significant difference. In total, 70 samples were only positive in at least two serologic tests and negative in all performed PCRs.

Titers judged positive >1:20 were diluted until a 1:1280 maximum. In most cases, the results showed highly positive titers (Figure 1).

## 6. PCR Run Analysis

Analysis with the 87 sequenced samples (please see also Table 3):

All PCRs were run in triplicate (same extraction sample used).

In the kDNA-RT-PCR (in total 225 positive results): Sixty-five samples were positive in three runs, 10 in two runs, and five in one run. Within the 87 sequenced samples, the kDNA-PCR produced 72 wrong results, mainly due to cross-reactivity with *T. rangeli* (20.5%).

Out of 71 samples being positive in at least one run and two serology results, 65 were sequenced as *T. cruzi*.

Out of 10 samples being positive in two serologies and the kDNA-PCR only, *H. sapiens* eight times, once for *T. rangeli*, and once for *T. cruzi* were sequenced.

The sensitivity for kDNA-PCR within the 87 sequenced samples was found to be 89.23%, and the specificity 22.73% (Table 3 and Table 4).

Regression analysis demonstrated that for kDNA-PCR, at least three runs are necessary to increase the reliability of the results, not considering cross-reactions (e.g., with *T. rangeli*), which cannot be eliminated by repetition.

The 18S rRNA-RT-PCR yielded three positive runs in total: No sample was positive in all three runs, one in two runs, and one in one run (two samples). At least one positive 18S rRNA-PCR run and two positive serologic results were found in both samples. According to the sequence analysis, it was *T. cruzi*. Cross-reactivity with *T. rangeli* or *H. sapiens* was not seen. The 18S rRNA-PCR showed an overall sensitivity of 1.53% and a specificity of 100%. For further information, please see Table 3 and Table 4.

For 18S rRNA-PCR, the regression analysis showed that at least 12 runs are necessary to overcome the small detection rate.

In TCZ-PCR (in total 40 positive results): Six samples were positive in all three runs, six in two runs, and 10 in one run. According to sequence analysis, it was *T. cruzi* in all cases. Cross-reactivity with *T. rangeli* or *H. sapiens* was not seen.

At least one positive TCZ-PCR run and two positive serologic results were found in all 21 samples. Notably, in one sample only, the TCZ-PCR (together with the NDO-PCR) revealed a positive result. Sequencing confirmed infection by *T. cruzi*.

The TCZ-PCR showed an overall sensitivity of 20.51% and a specificity of 100%. For further information, please see Table 3 and Table 4.

Regression analysis indicated, that at least six runs are necessary to obtain a reliable result.

In NDO-PCR (in total 180 positive results): Fifty-six samples were positive in all three runs, four in two runs, and four in one run. In all cases, sequencing revealed *T. cruzi* to be the underlying pathogen. In one sample, after sequencing, the kDNA-PCR amplicon revealed an infection with *T. cruzi* (serology positive), however, this was not detected by the NDO-PCR, therefore one false negative result needs to be registered. Cross-reactivity with *T. rangeli* or *H. sapiens* was not seen.

At least one positive NDO-PCR and two positive serologic results were found in 61 samples.

In one case, only the NDO-PCR provided a positive result (boy, three years old, Marocaso), for which sequencing revealed a *T. cruzi* infection. In one case, the NDO-PCR and TCZ-PCR showed a positive result (girl, two years old, Tezhumake), and *T. cruzi* was confirmed by the sequencing data. The NDO-PCR showed an overall sensitivity of 92.31% and a specificity of 100%. For further information, please see Table 3 and Table 4.

Regression analysis revealed that one run is sufficient to gain a reliable result.

### Analysis of the Non-Sequenced Samples (265)

From the 352 samples, in total 87 were sequenced and 265 were not sequenced. In the 265 not sequenced samples, another positive result was detected by the NDO-PCR only. The boy lives in the same household as the 2-year old positive girl (NDO- and TCZ-PCR positive only). In the kDNA-PCR, a further 27 positive results arose (combined with negative serologies and negative NDO- and other PCRs, indicated to be *T. rangeli*).

Forty-two cases were positive considering the results of all four PCRs, and 12 cases, if kDNA-PCR was subtracted.

Extrapolating the NDO-PCR results from the 87 sequenced samples and taking the NDO-PCR as the gold standard revealed similar outcomes as using the 87 sequenced samples alone. For further information, please see Table 3 and Table 4.

## 7. Disease Classification Outcomes

### 7.1. Sequenced Samples (87)

Negative in all serologic tests, but positive in at least one PCR run was true in 15 cases. Thirteen cases were false positive and only detected by the kDNA-PCR, where sequencing revealed *T. rangeli* in all cases.

One sample was positive in the NDO-PCR only and one in the NDO- and TCZ-PCR only. In both cases, a confirmatory sequencing revealed *T. cruzi* as the pathogen and were considered as acute *T. cruzi* infections. Age (three and two years), living place (simple housing with palm roof and mud walls), and endemic area support the findings.

Positive in at least two serologic tests and in at least one run of 18s rRNA-, TCZ- and/or NDO-PCR showed 61 positive cases. According to the definition, these could be considered as acute, re-infected, re-activated, and/or (early) chronic phases of CD. Considering the age of the volunteers, re-infection or pathogen detection within the chronic phase seems most likely. The number increases to 71 positive samples, if the analysis by kDNA-PCR is added and cross-reactivity disregarded.

Chronic cases (only positive in at least two serologic tests) were not present in the 87 sequenced samples as the sequencing needed PCR amplification products.

### 7.2. Not Sequenced Samples (265)

According to the NDO-PCR, one further acute case was detected. Following the results of the kDNA-PCR, a further 27 cases were acute.

Chronic cases were found in 70 cases.

Serology and PCR positive cases were found in 42 cases.

### 7.3. Overall Analysis (352)

Three acute cases, all detected by the NDO-PCR and one by NDO- and TCZ-PCR were found. kDNA-PCR showed 42 acute cases.

### 7.4. Chronic Cases Were Found in 70 Cases

Positive in two serologies and at least one PCR run was true for 113 samples, subtracting the kDNA-PCR in 103 cases.

## 8. Test Combinations

To evaluate the potential of test combinations (serology and PCR), the results of one or two positive serologic tests were linked to at least one positive run result of the PCRs. The data demonstrated that the use of one or two serologic tests did not show a significant difference. An improvement in diagnostic correctness could be seen for combinations of one (or two) serologic tests together with NDO-PCR, but not for kDNA-, TCZ-, and 18S rRNA-PCR. For further information, please see Table 5.

## 9. Discussion

A good tool, in particular, a gold standard, is urgently needed to provide a definite CD diagnosis. Due to a lack of reliable PCRs, diagnosis relies mainly on serology, accepting its limitations. One main limitation is the missing ability to detect acute cases, although the access to early diagnosis and care is essential, as early stages have the best outcomes for therapeutic success [11,14,37]. To improve the current situation, we developed a new Chagas-RT-PCR (NDO-PCR).

According to the WHO classification, the diagnosis is made during acute phase or re-activation by the direct detection of the parasite in the circulating bloodstream. Usually blood smear or concentration methods, followed by microscopic examination, are used. As the PCRs are also a direct method for the detection of the specific pathogen in blood, we claimed a positive PCR as an acute or re-activated CD.

The study was performed in a highly endemic CD region called La Guajira and Cesar in the northeast of Colombia. Two serologic tests were performed as recommended by the WHO. In addition, a Rapid Test, three leading PCRs, and the newly developed one (NDO-PCR) were applied.

During the study, we found inconsistent kDNA-, TCZ-, and 18S rRNA-PCR results. This became even more obvious when the samples were run in triplicate. To know the correct status of the volunteers, a pool of 87 sequence samples was sequenced by plasmid vector preparation. The 87 sequenced samples revealed an infection with *T. cruzi* in 65 cases, *T. rangeli* in 14 samples, and an unspecific human DNA, but not *Trypanosoma* in eight cases. Compared to the previously published sensitivity and specificity of kDNA-PCR (78%/40%) [15], an increase in the sensitivity and a decrease in specificity (89%/23%) were found. The decrease in specificity was mainly due to cross-reactions with the apathogenic *T. rangeli*, as already described previously [22]. The specificity of 100% in TCZ- and 18S rRNA-PCR was confirmed, however, the sensitivities (63% and 6%) [15] decreased markedly (20.5% and 1.5%) in the newly gained data.

The different outcomes can be explained by the different patient collectives analyzed. Qvarnstrom et al. [15] focused mainly on immune-compromised patients with CD, for whom high levels of parasitaemia in the blood can be expected, while the actual study analyzed samples from indigenous populations living in high-endemic areas for CD (natural rural setting) where low, moderate, and high pathogen levels are present. Despite this, the NDO-PCR showed the best sensitivities and specificities (92.3% and 100%). Moreover, the values for accuracy (kDNA-PCR 84.75%, 18S rRNA-PCR 78.88%, TCZ-PCR 83.71%, and NDO-PCR 98.39%), positive predictive value (59.6%, 75%, 93.4%m and NDO-PCR 100%), and negative predictive value (96.3%, 78.9%, 83.1%, and NDO-PCR 98%) showed superior outcomes for the NDO-PCR. However, even if the sensitivity and specificity of kDNA- and TCZ-PCR are considered as previously reported (e.g., by Qvarnstrom et al. (2012) with 78%/40% and 63%/100%, or Ramirez et al. (2015) with sensitivities of 84.14%/80.69% [15,21]), the performance of the NDO-PCR was superior (92.3%/100%). These results are due to the new primer and probe design, which is highly sensitive and specific to *T. cruzi*, not labeling *T. rangeli,* is able to detect *T. cruzi*, and is even one magnitude superior than the kDNA-PCR. With regard to kDNA-, TCZ-, and 18S rRNA-PCR, we agree with the conclusion that they cannot serve as a reliable CD detection tool [15,20], but with the NDO-PCR, a new and safe tool for the detection of *T. cruzi* is now available. It was the only PCR that reliably detected all acute cases. Of course, the sensitivity of TCZ- and 18s rRNA-PCR could be increased by performing repetitive runs (six and 12 runs would be necessary), but as this would be too cost and time intense, this option seems unrealistic. In contrast, for the kDNA-PCR, only three confirmatory runs would improve the outcome, but the leading problem with cross-reactivity cannot be solved by repetition. However, for the NDO-PCR, only one run is necessary to obtain a trustworthy result.

Within the 265 not sequenced samples, the TCZ- and 18S rRNA-PCRs showed only very few positive results in these high endemic settings. In contrast, the kDNA-PCRs indicated a high number of positive results, likely to be a mix of *T. rangeli* and *T. cruzi* infections. The NDO-PCR presented realistic results, fitting into the clinical and epidemiological setting and was consistent with the serologic data. The extrapolation of the NDO-PCR with the 265 not sequenced samples showed comparable results, confirming the NDO-PCR as a reliable tool.

The WHO recommends the use of two different serologic tests to diagnose CD. However, related to the well-defined 87 sequenced samples, serology gave 19% of false results. In addition, there was no significant difference in the outcome if the RT or one or two serologic tests were used. An increase in diagnostic safety by combining serology with kDNA-, TCZ-, or 18S rRNA-PCR could not be demonstrated. However, if at least one serologic test was combined with the NDO-PCR, the outcomes improved significantly. Not only were acute cases detected reliably, thus closing the gap of serologic tests, but a high number of (early) chronic cases could also be found, in which the pathogen was still circulating in the bloodstream.

If the therapy indication is to have two positive serologic tests, as recommended by the WHO, 183 within the study would be eligible for treatment. Seventy of those were just positive in two serologic tests, being negative in all PCRs and 113 were positive in at least one PCR run and two serologic tests, and 103 if kDNA-PCR was not considered (false positive results verified by sequencing subtracted).

In our setting, 70 volunteers were positive in two serologic tests and negative in all PCRs. None of them showed clinical signs or symptoms for CD. The question of whether these patients belong to the 30–40% of infected who will develop CD associated complications [9,14,15,38] and need treatment, or who rather belong to the 60–70% of infected who will cope well with the infection, is crucial. However, during the chronic indeterminate phase, no or low numbers of the pathogen can be found in the blood. Due to the remarkable detection rate of the NDO-PCR alone here in the study, 29% of the volunteers were positive in at least two serologic tests and one NDO-PCR run. It would be interesting to follow up on these groups to see whether during the course of the disease, different levels of pathogen circulate. It may be in those patients in whom later complications occur, low levels of parasites are detectable (permanently or undulating), while in those who cope the infections well, a chronic silent form is present.

The NDO-PCR could also give a hint of whether the pathogen is eliminated after treatment or remains below the detection rate, always ready to return. It is most important that the indication for therapy must be made carefully and restrictively [3,17,19,39] as it is a long-term treatment including possible severe side effects. Beyond this, only two drugs are available for which parasite resistance is feared.

Considering the clinical data, only 98 of the 352 participants stated complaints. On one hand, this is due to the cultural behavior of the indigenous population as it is not well accepted in the society to complain. On the other hand, most communities are about 6 h from the next health center, which is only sparsely equipped, and about 12 h walking distance from the next eligible hospital. Therefore, only long-lasting and as serious considered complaints are taken care of. As diarrhea, for example, is common in almost the whole community, it is considered as a minor problem or taken as normal status and therefore often not mentioned as conspicuous. Thus, the anamnestic assessment of the complaints has a lot of bias and is difficult to evaluate. However, the physiological findings in the doctor’s examination showed a remarkably high number of heart murmurs, being the leading finding (23.3%) accompanied with other cardiac problems (6.8%). This could be related to CD, but as, for example, systolic heart murmurs are also found in other diseases like anemia, it cannot be assigned to CD directly. Further diagnostics like electrocardiogram (ECG), heart echo, etc. would have been helpful, but were not available at that time. Additional examinations such as ECG, etc. and more supporting data would be favorable, gained by higher numbers of patients.

In Colombia, mainly type TcI of *T. cruzi* is prominent (74.2%), but TcII (17.2%), TcIII (1.48%), TcV (0.5%), and mixed infections (6.7%) are also present [40,41]. The NDO-PCR showed its ability to detect TcI and TcII, so far, therefore it is necessary to further evaluate whether the NDO-PCR also serves for the detection of other types/subtypes [22,42,43,44]. However, it has already shown its abilities for the most widespread type of *T. cruzi,* TcI, and can therefore be of good use for other affected countries.

In conclusion, the NDO-PCR is a novel Chagas-RT-PCR with the highest specificity, sensitivity, and accuracy when compared to existing PCR methods. In combination with a serologic test, it can significantly improve CD management. Furthermore, it can be used for therapy indication, monitoring, and control as well as for vector and surveillance control and public health purposes.

## Figures and Tables

**Figure 1 jcm-09-01517-f001:**
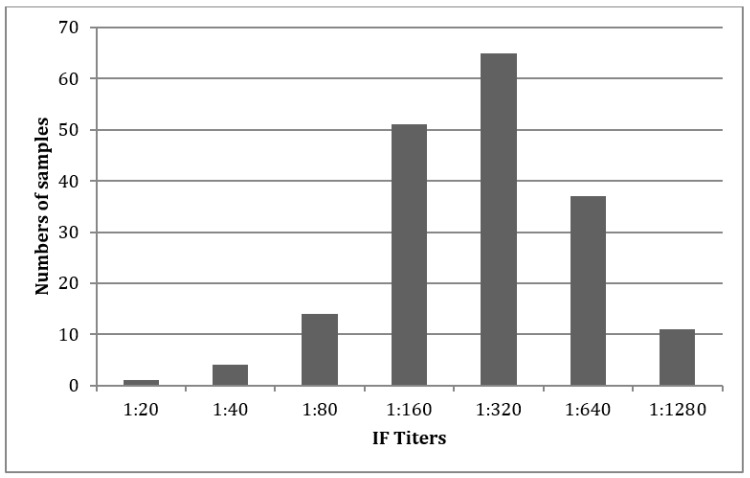
Immunofluorescence titer distribution in the analyzed samples.

**Table 1 jcm-09-01517-t001:** Demographic overview of the 352 samples. N = number.

Variable	Sequenced Samples (*n* = 87)	Unsequenced Samples (*n* = 265)	All Samples (*n* = 352)
**Age**			
N (%)	86 (98.85)	265 (100)	351 (99.72)
Mean	23.92	18.95	20.17
Median	16	12	14
**Gender (%)**			
Female	49 (56.32)	133 (50.19)	182 (51.70)
Male	38 (43.68)	132 (49.81)	170 (48.30)
**Region (%)**			
Wiwas	69 (79.31)	265 (100)	334 (94.89)
Wintukwa	18 (20.69)	0 (0)	18 (5.11)
Ashintukwa	9 (10.34)	22 (8.3)	31 (8.81)
Marocaso	16 (18.39)	69 (26.04)	85 (24.15)
Siminke	19 (21.84)	69 (26.04)	88 (25.00)
Tezumake	21 (24.14)	105 (39.62)	126 (35.80)
Savannah Crespo	4 (4.60)	0 (0)	4 (1.14)
Arwamake	17 (19.54)	0 (0)	17 (4.83)
Timaka	1 (1.15)	0 (0)	1 (0.28)

**Table 2 jcm-09-01517-t002:** Serologic test results. PPV: positive predictive value, NPV: negative predictive value, FNR: false negative rate, FPR: false positive rate.

Serology (At Least 2 Tests Positive)	87 Sequenced Samples *	265 Non-Sequenced Samples *	352 All Samples *
Positive result	71	112	183
Negative result	16	153	169
Sensitivity (%)	96.92		
Specificity (%)	59.09		
PPV (%)	87.50		
NPV (%)	86.67		
FNR (%)	3.08		
FPR (%)	40.91		
Accuracy (%)	87.36		

* All data in the table refer to the serology results in the specific groups.

**Table 3 jcm-09-01517-t003:** PCR results compared to the 87 sequenced samples. Positive was counted, if at least one PCR run (triplicate) was positive. PPV: positive predictive value, NPV: negative predictive value, FNR: false negative rate, FPR: false positive rate, ST: serologic test.

87 Sequenced Samples (Total Number of Test Runs: 261)	PCRs			
	kDNA	18S rRNA	TCZ	NDO
True positive	174	3	40	180
False positive	51	0	0	0
False negative	21	192	155	15
True negative	15	66	66	66
Sensitivity (%)	89.23	1.53	20.51	92.31
Specificity (%)	22.73	100	100	100
PPV (%)	77.33	100	100	100
NPV (%)	41.67	25.58	29.86	81.48
FNR (%)	10.77	98.46	79.49	7.69
FPR (%)	77.27	0	0	0
Accuracy (%)	72.41	26.44	40.61	94.25

**Table 4 jcm-09-01517-t004:** Extrapolation, taking the Newly Developed One Real-Time Polymerase Chain Reaction (NDO-RT-PCR) as a gold standard. Positive, if at least one PCR run (triplicate) was positive. PPV: positive predictive value, NPV: negative predictive value, FNR: false negative rate, FPR: false positive rate. NDO-PCR shows best sensitivity and specificities values compared to the other PCRs.

352 Samples (87 Sequenced, 265 not Sequenced), (total Number of Test Runs: 1056)	PCRs			
	kDNA	18S rRNA	TCZ	NDO
True positive	198	3	57	208
False positive	134	1	4	0
False negative	27	222	168	17
True negative	697	830	827	831
Sensitivity (%)	88.00	1.33	25.33	92.44
Specificity (%)	83.87	99.88	99.52	100
PPV (%)	59.64	75.00	93.44	100
NPV (%)	96.27	78.90	83.12	98.00
FNR (%)	12.00	98.67	74.67	7.56
FPR (%)	16.13	0.12	0.48	0
Accuracy (%)	84.75	78.88	83.71	98.39

**Table 5 jcm-09-01517-t005:** Comparison of different methods and combinations of methods related to the 87 sequenced samples (65 *T. cruzi*, 14 *T. rangeli*, 8 *H. sapiens*). ST: serologic test positive; * PCR positive in at least one run, n.a.: not applicable. One rapid test combined with NDO-PCR showed the best results.

Methods	No. of Positive Results	No. of True Positive Results	No. of False Positive Results	Missed Chronic *T. cruzi* Infections	Missed Acute *T. cruzi* Infections
2 ST only	71	62	9	1	n.a.
1 ST + kDNA-PCR *	72	63	9	0	2
1 ST + 18S rRNA-PCR *	2	2	0	61	2
1 ST + TCZ-PCR *	21	21	0	42	2
1 ST + NDO-PCR *	62	62	0	1	2
*2 ST + kDNA-PCR **	71	62	9	1	2
*2 ST + 18S rRNA-PCR **	2	2	0	61	2
*2 ST + TCZ-PCR **	21	21	0	42	2
*2 ST + NDO-PCR **	61	61	0	2	2
0 ST + kDNA-PCR *	13	0	13	n.a.	2
0 ST + 18S rRNA-PCR *	0	0	0	n.a.	2
0 ST + TCZ-PCR *	1	1	0	n.a.	1
0 ST + NDO-PCR *	2	2	0	n.a.	0

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
