# Peer review of "Chagas Disease: Detection of *Trypanosoma cruzi* by a New, High-Specific Real Time PCR"

_jcm, 2020, doi:10.3390/jcm9051517_

Round 1

Reviewer 1 Report

The authors have addressed the concerns raised during my initial reviews. No further comments from me.

Author Response

Reviewer 1: The authors have addressed the concerns raised during my initial reviews. No further comments from me.

We thank the Reviewer for accepting our paper.

Reviewer 2 Report

The manuscript by Kann et al. describes the development of a novel nucleic acid amplification for the detection of Trypanosoma cruzi infections. The manuscript is very poorly put together and requires extensive editing for syntax.

For example (only some are listed):

Line 80

Line 124

Line 127-128

Line 138

Line 238

Minor Points:

Line 59: Chagas is endemic to the US. Authors need to distinguish between endemic areas and regions where infected individuals travel to.

Line 85: What do the authors mean by ‘seroconversion after treatment is rare…’

Line 115: Why is the study focused on individuals with Dengue fever and fever of unknown origin and why is ‘Fever of Unknown Origin’ capitalized?

Lines 284-288: Why is each sentence in an individual line? It is not clear of the test described on line 285 and the test described on line 286 is the same sample.

Major Points:

Lines 149-150: Do the authors mean that participant serum was used to detect fixed promastigotes? Include relevant reference.

Lines 174-195: This is the Sequencing section of the Methods. Most of what is written here belongs in the Results section.

Line 229: Why is there is Disease Classification section in the Methods?

A demographics table of the population enrolled is missing, that is, age, gender….

Line 313, 338-346: Are the data reference on this line present on Table 2? Indicate Table 2 in the text if yes.

Table 2: list the total number of tests run.

Line 358: The one ‘positive sample that was not detected…’ how do the authors know it was positive?

Author Response

Reviewer 2:

The manuscript by Kann et al. describes the development of a novel nucleic acid amplification for the detection of Trypanosoma cruzi infections.

We thank the reviewer for observing this.

The manuscript is very poorly put together and requires extensive editing for syntax.

Here the reviewer comments were actually quite helpful and we acknowledge the time invested by the reviewer to point out here details, very helpful.

For example (only some are listed):

Sentences were rephrased throughout the manuscript and the manuscript polished. The new wording for the examples given is now:

Line 80 ? „As they are mainly antibody based and the antibody production needs several weeks, the patient has already entered the chronic phase when the serologic tests turn positive.“

Line 124 ?? „With regard to the indigenous populations, all inhabitants of the villages were invited to participate.“

Line 127-128 „Volunteers aged 12 years and above received a CD rapid test (RT) and a blood withdrawal, if the RT was positive. The serum was then analyzed further with ELISA/ Immunofluorescence (IF) and the 4 different PCRs.“

Line 138 „In 2014, a retrospective study was performed, using 352 serum samples collected from indigenous populations living in CD high-endemic regions of César (Tezhumake) and La Guajira (Arwamake, Timake, Marocaso, Siminke, Ashintukwa), in the northeast of Colombia.“

Line 238 „Here a Rapid Test and ELISA/IF was performed, as requested by the Colombian guidelines, to entitle the patient for treatment.“

Minor Points:

Line 59: Chagas is endemic to the US. Authors need to distinguish between endemic areas and regions where infected individuals travel to.

 The sentence was clarified: „Although prevention, control measures and overall improvements of life quality led to a decrease of incidence and prevalence of CD, it remains a major threat in Latin America. It is endemic in Mexico, in the central part and the southern half of the United States [5] and expands more and more to non-endemic countries such as Canada and Europe, mainly due to migration and travelling [6-12].“

Line 85: What do the authors mean by ‘seroconversion after treatment is rare…’

This was explained more clearly: „ Furthermore, as Chagas antibodies can remain positive for years, for decades or even for a lifetime, and a new negative serologic test after infection and/or treatment is rare, it is difficult to assess the therapy outcome [16-18].“

Line 115: Why is the study focused on individuals with Dengue fever and fever of unknown origin and why is ‘Fever of Unknown Origin’ capitalized?

These statements are now further polished and explained in the paper: „The study focused on Dengue Fever and fever of unknown origin. The intention was to identify the underlying diseases. Especially in regions of Colombia, the etiology and epidemiological characterization of those is difficult [22]. During the study we accessed also the regions of the indigenous populations, where fever-associated diseases are very common. Here we saw that CD was one of the main problems.“

Lines 284-288: Why is each sentence in an individual line? It is not clear of the test described on line 285 and the test described on line 286 is the same sample.

The Paragraph is now changed to: „According to max score, total score, query cover and expected value, the blasted sequences revealed 63 samples of T. cruzi, 14 of T. rangeli and 8 cases of Homo sapiens (unspecific human DNA, not Trypanosoma). The average identification correctness was 91.4%. In one sample, only the TCZ-PCR (and later also the NDO-PCR) showed a positive result. Also this TCZ-PCR amplicon was sequenced as described above and revealed an infection with T. cruzi.

After the development of the NDO-PCR, another case occurred in which only the NDO-PCR showed a positive result. Sequencing of the NDO-PCR amplicon, performed as described above, presented T. cruzi as underlying pathogen.

In total, 87 samples were successfully sequenced and 65 were defined as T. cruzi, 14 as T. rangeli and 8 as H. sapiens (unspecific human DNA, not Trypanosoma). The NDO-PCR and the other leading PCRs were compared to this data set, which was further used as gold standard.”

Major Points:

Lines 149-150: Do the authors mean that participant serum was used to detect fixed promastigotes? Include relevant reference.

Yes, we used indirect immunofluorescence on fixed promastigotes. This is an in-house test, which is a certified test of the Bernhard-Nocht Institute, Germany´s largest tropical institute, and is validated regularly in official tests series to monitor proficiency.

The sentences was rephrased to: “Positive RTs and/or ELISAs were confirmed by Indirect Immunofluorescence (IF) on fixed promastigotes of T. cruzi (Tulahuen strain TcII, from cell cultures) according to standard procedures [24].”

Lines 174-195: This is the Sequencing section of the Methods. Most of what is written here belongs in the Results section.

We thank the reviewer for this suggestion. The section was restructured. The description of the method was shortened and the results parts placed into the results section.

Line 229: Why is there is Disease Classification section in the Methods?

Thank you for this suggestion, we put this section now into the results part.

A demographics table of the population enrolled is missing, that is, age, gender….

We added now a demographic table briefly summarizing the population enrolled as requested.

Table 1: Demographic overview of the 352 samples. N=number; Nmiss=number of missing-values

Variable

Sequenced samples (N=87)

Unsequenced samples (N=265)

All samples (N=352)

Age

N (Nmiss)

86 (1)

265 (0)

351 (1)

Mean

23.92

18.95

20.17

Median

16

12

14

Gender (%)

Female

49 (56.32)

133 (50.19)

182 (51.70)

Male

38 (43.68)

132 (49.81)

170 (48.30)

Region (%)

Wiwas

69 (79.31)

265 (100)

334 (94.89)

Wintukwa

18 (20.69)

0 (0)

18 (5.11)

Ashintukwa

9 (10.34)

22 (8.3)

31 (8.81)

Marocaso

16 (18.39)

69 (26.04)

85 (24.15)

Siminke

19 (21.84)

69 (26.04)

88 (25.00)

Tezhumake

21 (24.14)

105 (39.62)

126 (35.80)

Savannah Crespo

4 (4.60)

0 (0)

4 (1.14)

Arwamake

17 (19.54)

0 (0)

17 (4.83)

Timake

1 (1.15)

0 (0)

1 (0.28)

Line 313, 338-346: Are the data reference on this line present on Table 2? Indicate Table 2 in the text if yes.

This information was added directly after the headline “Analysis with the 87 sequenced samples”. The re-numeration was adapted.

Table 2: list the total number of tests run.

The total number of test runs was added.

Line 358: The one ‘positive sample that was not detected…’ how do the authors know it was positive?

This was clarified in the following sentence: „In one sample, the kDNA-PCR amplicon revealed after sequencing an infection with T. cruzi (serology positive), this however was not detected by the NDO-PCR, therefore one false negative result needs to be registered.“

Reviewer 3 Report

In this work, Kann et al reported the development of a new molecular test (PCR-based) to diagnose Trypanosoma cruzi in dwellers from a Caribean region considered as endemic for Chagas disease. The samples size was appropriated, and the methodology was properly used with rigour in the controls implemented for comparisons. The newly developed rt-PCR method enabled the detection with an overall 94% accuracy, 92% of sensitivity and it was 100% specific for T. cruzi. The detection limit was 1,5 copies/µl. There are few suggestions in the way data were presented in order to facilitate the delivering of the message and that would favour data interpretation.  

  • Information presented at the methodology section (line 181-195) refers to method development, so it would be better to move to the results section. 
  • Paragraph (Line 214-223) should be moved as suggested above.
  • typos: line 291 a dot can be deleted, line 461: kDNA-PCR. 
  • please clarify if PCR ran in triplicates means the samples were extracted 3x separately and then used as template in PCR test? or the same DNA sample was used in three different runs?
  • line 315: please specify what do you mean by saying 'wrong results'? If there was cross-amplification with T. rangeli sequences then, sequences alignment should be included indicating the undesired region amplified.

Author Response

Reviewer 3

In this work, Kann et al reported the development of a new molecular test (PCR-based) to diagnose Trypanosoma cruzi in dwellers from a Caribean region considered as endemic for Chagas disease. The samples size was appropriated, and the methodology was properly used with rigour in the controls implemented for comparisons. The newly developed rt-PCR method enabled the detection with an overall 94% accuracy, 92% of sensitivity and it was 100% specific for Tcruzi. The detection limit was 1,5 copies/µl.

We thank the Reviewer for summarizing our main results.

There are few suggestions in the way data were presented in order to facilitate the delivering of the message and that would favour data interpretation.  

  • Information presented at the methodology section (line 181-195) refers to method development, so it would be better to move to the results section. 

Yes, we transferred this information now into the results section.

  • Paragraph (Line 214-223) should be moved as suggested above.

This was moved as suggested.

  • typos: line 291 a dot can be deleted, line 461: kDNA-PCR. 

Thanks, this was corrected.

  • please clarify if PCR ran in triplicates means the samples were extracted 3x separately and then used as template in PCR test? or the same DNA sample was used in three different runs?

The same extraction sample was used three times, otherwise the costs for extraction kit, personal and time would have been too high. This is now also clarified in the methods.

  • line 315: please specify what do you mean by saying 'wrong results'? If there was cross-amplification with Trangelisequences then, sequences alignment should be included indicating the undesired region amplified.

The sequence analysis revealed in these cases the sequence for T. rangeli and not T. cruzi, therefore we named them as false positive or wrong results, as the PCRs are supposed to just detect T. cruzi.

We found TrSA7 to be the most often found T. rangeli strain. This additional information was added in the text in various places to clarify.

Reviewer 4 Report

Manuscript Number: jcm-763587                   Kann et al.                               30th March 2020

Major revision

This is an interesting paper, but presented as a confused draft. M&M, Results and Discussion are mixed, e.g. lines 191-192 are Results and presented in M&M; lines 229-234 should be presented in the Discussion, not in M&M; lines 264-280 should be placed into M&M, not in the Results. In all tables, the authors should delete the border lines and all horizontal lines below the heading line.

In the following, I include comments which refer to the respective line.

42: The authors should include a recent estimation of WHO.

52: The authors should write the species name in italics.

99: Cross-reactions with Leishmania also occur.

143: There are strict regulations to transport infectious material by plane. This should be included.

150: The authors should add a citation. In addition, they should specify the origin of the stages of T. cruzi. Did they use trypomastigotes (from blood, in vitro cultures, cell cultures) or epimastigotes (from the vector or cultures)? T. cruzi rarely develops promastigotes and then only as short-term intermediate stages.

174: The authors should replace “showed different results” by “differed”.

216-218: The authors should replace “Cl. Brenner” by “CL Brener” and “T. cruzi Typ Y” by “Y”. In addition, they should not write names of strains and spp. in italics. They should also add the DTU after the names of the strains (also for Tulahuen) and specify which species of Leishmania and which strain of T. rangeli they used (in the latter also the classification). I found no T. cruzi strain Brazil. Therefore, the authors should include the origin. So far, they only tested against strains belonging to two DTUs.

305-306: The authors should delete “An overview of the results can be found in Figure 1.” And place the indication to the figure in brackets at the end of the preceding sentence.

323: see 305-306

324, 333: The authors should correct the height of the lines in the last column.

389: The authors should correct the typing mistake.

531: The authors should include the other DTUs present there.

574: The authors should include a citation which can be followed.

578-796: The references need a correction according to the Instructions. I include only two general mistakes.

      Species names should be typed in italics.

      English titles should be written in lower case.

Author Response

Reviewer 4

This is an interesting paper, but presented as a confused draft. M&M, Results and Discussion are mixed, e.g. lines 191-192 are Results and presented in M&M; lines 229-234 should be presented in the Discussion, not in M&M; lines 264-280 should be placed into M&M, not in the Results. In all tables, the authors should delete the border lines and all horizontal lines below the heading line.

We thank the reviewer for his helpful recommendations and lauding the paper. The restructuring was done according to the recommendations given.

In the following, I include comments which refer to the respective line.

42: The authors should include a recent estimation of WHO.

This was included as suggested: „It is a major burden in Latin America with about 8 million infected patients according to recent estimates by WHO [1], concurrently geographically expanding more and more.“

52: The authors should write the species name in italics.

This was changed as suggested.

99: Cross-reactions with Leishmania also occur.

Right, this was added as requested.

143: There are strict regulations to transport infectious material by plane. This should be included.

The procedure following the appropriate regulations was included. It is now: „International guidelines for airfreight sample transport were followed, permissions gained from all involved parties, a permanent cooling chain was controlled and confirmed. After the arrival in Germany, the sera and extracts were stored at -80°C until their further use for analysis.“

150: The authors should add a citation. In addition, they should specify the origin of the stages of T. cruzi. Did they use trypomastigotes (from blood, in vitro cultures, cell cultures) or epimastigotes (from the vector or cultures)? T. cruzi rarely develops promastigotes and then only as short-term intermediate stages.

Yes, we used indirect immunofluorescence on fixed promastigotes. This is an in-house test, which is a certified test of the Bernhard-Nocht Institute, Germany´s largest tropical institute, and is validated regularly in official tests series to monitor proficiency.

The sentences was rephrased to: “Positive RTs and/or ELISAs were confirmed by Indirect Immunofluorescence (IF) on fixed promastigotes of T. cruzi (Tulahuen strain TcII, from cell cultures) according to standard procedures [24].”

174: The authors should replace “showed different results” by “differed”.

This passage was rearranged, as also Reviewer 2 commented on this.

216-218: The authors should replace “Cl. Brenner” by “CL Brener” and “T. cruzi Typ Y” by “Y”. In addition, they should not write names of strains and spp. in italics. They should also add the DTU after the names of the strains (also for Tulahuen) and specify which species of Leishmania and which strain of T. rangeli they used (in the latter also the classification). I found no T. cruzi strain Brazil. Therefore, the authors should include the origin. So far, they only tested against strains belonging to two DTUs.

The section was done new, it is now: „Primer and probes were tested with known strains such as CL. Brener (DTU TcII), Y (DTU TcII) and Brazil (DTU TcI) [26-30] and showed the expected positive outcomes. Malaria spp., Leishmania brasiliense and T. rangeli (TrSA7) was used for cross-reactivity testing, which did not occur. For detection limit determination, DNA from the positive control Tulahuen (DTU TcII) [31-33], taken from tissue culture, was determined in a serial dilution. The lowest dilution was defined valid when 3/3 results were positive. The detection limit (LOD) of the NDO was compared to the kDNA-PCR one magnitude superior (1:100.000 versus 1:10.000; 1,5 copies/µl versus 15 copies/µl).“

305-306: The authors should delete “An overview of the results can be found in Figure 1.” And place the indication to the figure in brackets at the end of the preceding sentence.

This was done as requested.

323: see 305-306

This was done, too.

324, 333: The authors should correct the height of the lines in the last column.

The layout was changed completely according to the recommendations of Reviewer 2.

389: The authors should correct the typing mistake.

Thank you, the typing mistake was corrected.

531: The authors should include the other DTUs present there.

The information was added, it is now: „In Colombia, mainly type TcI of T. cruzi is prominent (74,2%), but also TcII (17,2%), TcIII (1,48%), TcV (0,5%) and mixed infections (6,7%) are present [37,38]. The NDO-PCR showed its ability to detect TcI and TcII so far, therefore it is necessary to evaluate further, whether the NDO-PCR also serves for the detection of the other types/subtypes [21,39-41]. However, it has already shown its abilities for the most spread type of T. cruzi, TcI, and can therefore be of good use for other affected countries.“

574: The authors should include a citation which can be followed.

Thank you for this hint, the citation was corrected.

578-796: The references need a correction according to the Instructions. I include only two general mistakes.

      Species names should be typed in italics.

      English titles should be written in lower case.

The reference format was updated as suggested.

Round 2

Reviewer 4 Report

Manuscript Number: jcm-763587 v2             Kann et al.                               2nd May 2020

Major revision

The revision has strongly improved the manuscript:

In the following, I include comments which refer to the respective line. Those marked in bold are insufficient corrections of the previous draft.

42, 63: The authors should include a recent estimation of WHO. An estimation of 2007 isn´t recent!!

59: The authors should re-write this sentence. Mexico belongs to Latin America.

144: The authors should replace “male” by “from male”.

145: The authors should delete the information in brackets (repetition).

148-149: The authors should present the number of the missing value in square brackets and include that they present percentages in brackets.

182: T. cruzi rarely develops promastigotes and then only as short-term intermediate stages. Therefore, the authors presumably used trypomastigotes!!! The first author should contact Prof. Fleischer at the BNI.

318-334: This part belongs to M&M.

338: The authors should replace “dilutions.” by “dilutions and the”.

339-341: The authors should include these parasites in M&M (plus origin, cultivation conditions, stage, DNA-extraction).

341: The authors should not write spp. in italics.

358: The authors should not write “(unspecific human DNA, not” in italics.

365-366: The authors should delete this sentence (repetition with only a slight difference).

373, 551: The authors should not write “(unspecific DNA” in italics.

457: The authors should delete “in”.

667-800: The references need a correction according to the Instructions. I include only two general mistakes.

      Species names should be typed in italics, the genus name be typed in uppercase (Trypanosoma). My previous suggestion to use lower case is not relevant for countries, Chagas, PCR etc. Each citation has to be read carefully.

      Abbreviated journal name.

Author Response

In the following, I include comments which refer to the respective line. Those marked in bold are insufficient corrections of the previous draft.

42, 63: The authors should include a recent estimation of WHO. An estimation of 2007 isn´t recent!!

We thank the reviewer for this advice. The estimation was updated with data from WHO 2019. All passages affected were adjusted.

59: The authors should re-write this sentence. Mexico belongs to Latin America.

We thank the reviewer for this hint. The sentence was changed to: „Above, it is endemic in the central part and the southern half of the United States [5] and expands more and more to non-endemic countries such as Canada and Europe, mainly due to migration and travelling [6-11] [12].“

144: The authors should replace “male” by “from male”.

This was changed according to the suggestion of the reviewer.

145: The authors should delete the information in brackets (repetition).

This is true, it was deleted.

148-149: The authors should present the number of the missing value in square brackets and include that they present percentages in brackets.

The graph was adjusted according to the recommendations.

182: T. cruzi rarely develops promastigotes and then only as short-term intermediate stages. Therefore, the authors presumably used trypomastigotes!!! The first author should contact Prof. Fleischer at the BNI.

We contacted Prof. Fleischer to verify again the stage of T. cruzi in the test. As a matter of fact he wrote to me, that the BNITM used epimastigotes for the test. We apologize for this mistake and are grateful for this valuable hint. Of course this was corrected.

318-334: This part belongs to M&M.

The part was moved to M&M.

338: The authors should replace “dilutions.” by “dilutions and the”.

This was done.

339-341: The authors should include these parasites in M&M (plus origin, cultivation conditions, stage, DNA-extraction).

We thank the reviewer for this comment and contacted Prof. Fleischer, from whom we obtained the materials. We added the following information to the paper:

“The three reference strains Cl Brener, Y and Brazil were obtained from the strain collection of the National Reference Center for Tropical Agents at the Bernhard-Nocht Institute for Tropical Medicine (BNITM) in Hamburg. The trypanosomes were passaged once in BALB/c mice and were then frozen in aliquots. Aliquots were thawed and propagated in Nakamura medium as epimastigotes. The Malaria spp. and Leishmania brasiliense samples were from travelers consulting the BNITM, returning from Ghana and Brazil. T. rangeli (TrSA7) was taken, after being analyzed by sequencing, out of the gained sample collection of the study. All extractions were performed as described above.”

341: The authors should not write spp. in italics.

Thank you for this hint, it was corrected.

358: The authors should not write “(unspecific human DNA, not” in italics.

This was corrected.

365-366: The authors should delete this sentence (repetition with only a slight difference).

The sentence was deleted.

373, 551: The authors should not write “(unspecific DNA” in italics.

This was corrected.

457: The authors should delete “in”.

This was also corrected.

667-800: The references need a correction according to the Instructions. I include only two general mistakes.

      Species names should be typed in italics, the genus name be typed in uppercase (Trypanosoma). My previous suggestion to use lower case is not relevant for countries, Chagas, PCR etc. Each citation has to be read carefully.

      Abbreviated journal name.

We thank the reviewer for carefully checking the references. However, we used the MDPI citation style in the Endnote citation manager, so that the references are automatically formatted according to the journal requirements. We will bring this point to the editorial office to check if additional format changes which are currently not implemented in the MDPI journal reference style are required."

This manuscript is a resubmission of an earlier submission. The following is a list of the peer review reports and author responses from that submission.

Round 1

Reviewer 1 Report

It’s reassuring that the authors clarify that the primers used for TCZ PCR where indeed correct and the inconsistencies in primer sequence with the traditional method were just typos. Nonetheless, the arguments the authors make in justifying the low sensitivity of TCZ in their hands does not solve the problem. As another example, a highly cited international study evaluated many PCR methods and found that TCZ and kDNA PCRs have similar sensitivities in serial dilutions of T. cruzi DNA and in clinical samples when performed by different labs throughout the world (Schijman, Alejandro G., et al. "International study to evaluate PCR methods for detection of Trypanosoma cruzi DNA in blood samples from Chagas disease patients." PLoS neglected tropical diseases 5.1 (2011): e931.). In fact, in this aforementioned study both assays have sensitivities in the fg/ul range, while the authors report that NDO-PCR has a sensitivity of 1 pg/ul. Therefore, the improvements in sensitivity in NDO-PCR over traditional methods with high specificity are most likely marginal. As previously stated, other highly sensitive and specific qPCR methods have demonstrated to be useful not only for detection, but also for quantification of parasite loads in clinical settings, while incorporating internal standards for the identification false positives (Duffy et al. 2009, Ramirez et al. 2015, ). These benefits outweigh small improvements in sensitivity. Altogether, the current study by Kann et al. lacks the novelty and impact needed to be published in the Journal of Clinical Medicine.

Other concerns:

The alignment provided as supp. Information should be in higher resolution and should include the regions of both primers and the probe.

Author Response

It’s reassuring that the authors clarify that the primers used for TCZ PCR where indeed correct and the inconsistencies in primer sequence with the traditional method were just typos. Nonetheless, the arguments the authors make in justifying the low sensitivity of TCZ in their hands does not solve the problem. As another example, a highly cited international study evaluated many PCR methods and found that TCZ and kDNA PCRs have similar sensitivities in serial dilutions of T. cruzi DNA and in clinical samples when performed by different labs throughout the world (Schijman, Alejandro G., et al. "International study to evaluate PCR methods for detection of Trypanosoma cruzi DNA in blood samples from Chagas disease patients." PLoS neglected tropical diseases 5.1 (2011): e931.). In fact, in this aforementioned study both assays have sensitivities in the fg/ul range, while the authors report that NDO-PCR has a sensitivity of 1 pg/ul. Therefore, the improvements in sensitivity in NDO-PCR over traditional methods with high specificity are most likely marginal. 

At first we want to acknowledge, that the reviewer accepts our apology for the typing errors. However, we cannot agree to his point about the sensitivities in regard to TCZ- and kDNA-PCR, nor in regard to the NDO-PCR.

Qvarnstrom et al published 2012 that the “analytical sensitivity for DTU I“ was 0.1 fg/µl and for DTU IV 1 fg/µl for TCZ and kDNA-PCR. This already demonstrates very nicely the variability between the detection of two different DTUs. How much more are the results incomparable, if different T. cruzi DNA strains are taken? They describe a 10f old serial dilution of T. cruzi DNA, taken from “epimastigote cells grown in LIT medium” of different stocks (Silvio X10, Cl. Brener, CANIII), but we used T. cruzi DNA from Tulahuen. Using of these different stocks could already be reason enough for the different outcome, but cannot be interpreted as an inferiority of the NDO against TCZ and/or kDNA-PCR. A direct comparison implies same conditions.

Moreover it is questionable, if the same cut off for the limit of detection was taken. The reviewer requests 3/3 runs to be positive and this works 2 dilution steps longer with the NDO-PCR than with the kDNA-PCR. In the Schijman paper it is not mentioned if the same criteria were used.

The reviewer also linked his results to the Nanodrop measurement. Concerning this statement and the detection limit of fg/µl, we are very critical. Originally, we didn´t use this method, as we favor new standards, but to fulfill this requirement we did. And also, here we found one magnitude of superiority of the NDO against the kDNA-PCR (and if the kDNA is equal to the TCZ, this in turn means also against the TCZ). However, we do not agree with the thinking, that reaching the femtogram zone is a sign of quality and would indicate a better sensitivity This conclusion cannot be drawn as the conditions for a comparison were not equal. The most objective result is given when direct comparison using the same conditions is made. Please find hereto attached our serial dilution experiment (attachment 1). As you can see, we did also reach the femtogram zone, but we did not consider it. However, at first the kDNA starts to become unreliable and at really low concentrations also the more sensitive NDO does. The reason therefore is the common knowledge, that with the grade of dilution the reliability and accuracy suffers and for clinical purposes, ranges of femtogram are without any meaning. The companies themselves just claim the Nanodrop system reliable in the range of 3700 ng/µl – 2 ng/µl.

It is one thing to be as good as the kDNA in the serial dilution, but another to persist in the field. As mentioned before, the analyzed material taken in the publication of Qvarnstrom were “DNA specimens from genetically distinct cultured T. cruzi strains plus blood specimen from chronically infected patients”. The 119 blood specimen were from patients living in the USA. At least 75/119 will have shown high levels of parasitemia, as they were either immunosuppressed (after heart transplant, AIDS), or suffer from acute CD infection (congenital, etc.). Still, in this favorable setting the authors claim: “ … the same PCR assay can perform differently depending on the geographic origin of the specimen” and “The kDNA TaqMan assay seems to be the most sensitive assay but it can amplify non-T.cruzi DNA, e.g. T. rangeli, and thus lead to false positives. The TCZ TaqMan assay has better specificity but as shown in this study can produce false positive PCR results as well”. It lists the sensitivities in a table, saying TCZ 63% and kDNA 78%. These 15% difference is not a slightly higher difference, besides, shows clearly a difference.

Also in the publication of Schijman the result is:” However, Sat-DNA (what is the TCZ) tests were less sensitive than kDNA-PCR tests to detect T. cruzi I DNA.” This alone shows, that the statement, that the TCZ is equal in sensitivity to the kDNA is wrong.

Summarizing the last comment of us, the TCZ shows fluctuations between 81% sensitivity (Ramirez), 50% in adults and 76% in infants in the study of Cura et al and 63% in the publication of Qvarnstrom. It has to be considered here, that all patient cohorts were either immunosuppressed or -compromised, congenital or acute infections, and only little blood samples were taken from a natural setting.

As previously stated, other highly sensitive and specific qPCR methods have demonstrated to be useful not only for detection, but also for quantification of parasite loads in clinical settings, while incorporating internal standards for the identification false positives (Duffy et al. 2009, Ramirez et al. 2015, ). These benefits outweigh small improvements in sensitivity. Altogether, the current study by Kann et al. lacks the novelty and impact needed to be published in the Journal of Clinical Medicine.

Of course other PCRs are available, but if performed in a natural setting and compared to our NDO, they lack either sensitivity and/or specificity. We calculated, that three runs would be needed with the kDNA PCR to get as reliable results for CD diagnose than with the NDO. For the TCZ it would even have to be 6, and 12 for the 18s rRNA. The NDO combines a good sensitivity and specificity for the first time and needs therefore to be published. Direct comparison gives here the proof! To not publish would mean to five away a good chance for many patients and would reject major advances in sensitivity (one order of magnitude) and specificity (important in clinic and for screening programs).

Other concerns:

The alignment provided as supp. Information should be in higher resolution and should include the.regions of both primers and the probe.

We thank the reviewer for this suggestion. The alignment was sharpened and is now in the new form in the supplemental information. Moreover, the sequences are given in full in the supplementary material part again. However, unfortunately the program allows no further marking.

Reviewer 2 Report

Manuscript jcm-704011

Kann & al. study is focused on Dengue Fever and Fever of Unknown Origin. The authors describe a new diagnostic test for the diagnosis and monitoring of Chagas Disease.

Abstract. According to the Journal Instruction for Authors, “the abstract should be a single paragraph and should follow the style of structured abstracts, but without headings”.

Authors Summary. Each statement should be followed by a reference. E.g. L44-47.

Introduction. More references to the factors connected to the variability of the PCR specificity/sensitivity would be useful to highlight the interest in the development of new diagnostic tests. Other tests available (ELISA, IF) are not very well explained for the point of view of their limitations. It is just a personal opinion, but these tests were used in the study, and deep comparison of the available diagnostic tests would underline the aim of the study. Because in the Study Design section the authors write that “The study focused on Dengue Fever and Fever of Unknown Origin” (L109), some references about the Fever of Unknown Origin could be added to explain the interest for the present study. The T.cruzi genetic heterogeneity could be explained from the beginning, to link with the conclusion of the study (L529).

L90-93 – the sentence is too long, a rephrase could be clarified what the authors said “… these PCRs ….. are limited in their reproducibility, and can, therefore, not be used as a reliable diagnostic tool” (ref 13, 17, 18). Someone might understand that PCRs are not recommended to be used at all. But the conclusions of the cited articles are quite different. The reference 13 (DOI: 10.1371/journal.pntd.0001689) conclusion is “These data strongly indicate that at least two PCR assays with different performances should be combined to increase the accuracy. This evaluation also highlights the benefit of extracting DNA from the blood specimen's buffy coat to increase the sensitivity of PCR analysis”.  The reference 17 (DOI: 10.1371/journal.pntd.0000931) “This study represents a first crucial step towards international validation of PCR procedures for detection of T. cruzi in human blood samples”. The reference 18 (DOI: 10.1016/j.jmoldx.2015.04.010) “This effort is a major step toward international validation of qPCR methods for the quantification of T. cruzi DNA in human blood samples…”. More in L152-153, the authors write “The TCZ- and 18S rRNA-RT-PCR were chosen due to their results in specificity (100%) (13, 17)”.

Materials and Methods. The inclusion /exclusion criteria are not very well synthesized.

L115 “healthy considered/asymptomatic persons”  means healthy considered or asymptomatic persons? I asked because, at least for me, it is not easy to link this information with the details from Clinical assessment (L263-273). Are these persons those 157 patients? (L272-273).  Further analysis of clinical data with tests performed in the present study would clarify the outcome of the study and clarify the authors’ discussions. The same, the 87 samples sequenced are from which persons, from a clinical point of view? (see L510, also) Therefore, clear inclusion/exclusion criteria are needed.  In the discussion section (L446-447) there is mention that “samples from the indigenous population” are tested. The explanations from L492-496 should be reflected in the Results section, also.

The reference for the WHO classification (L222-223) and for the Columbian guidelines (L230)

Results. Serology. L283-290. So, false-positive are 9 cases, and 2 false negatives?

L294- the same numbers of samples false positive in one serologic test and in two serologic tests?

Please, rephrase the sentences from L314-317 for clarity.

Table 1 – L324 ST: serologic test in the heading, but in the table, there are not data from the serologic tests.

References – please write the according to the JCM requirements – “In the text, reference numbers should be placed in square brackets [ ], and placed before the punctuation…”.

Author Response

Kann & al. study is focused on Dengue Fever and Fever of Unknown Origin. The authors describe a new diagnostic test for the diagnosis and monitoring of Chagas Disease.

Abstract. According to the Journal Instruction for Authors, “the abstract should be a single paragraph and should follow the style of structured abstracts, but without headings”.

Thank you for this advice. The abstract was changed according to the requirements and is now without headings and a single paragraph.

Authors Summary. Each statement should be followed by a reference. E.g. L44-47.

References were included.

Introduction. More references to the factors connected to the variability of the PCR specificity/sensitivity would be useful to highlight the interest in the development of new diagnostic tests.

Suitable references were added.

Other tests available (ELISA, IF) are not very well explained for the point of view of their limitations. It is just a personal opinion, but these tests were used in the study, and deep comparison of the available diagnostic tests would underline the aim of the study.

Thank you for this advice. To explain better the other available tests, the passage was changed to: “Although serologic tests serve well for the detection of chronic CD (2), e.g. ELISA, Immunofluorescence, rapid tests, etc., they have their limitations. As they are mainly antibody based and the infected body needs a couple of weeks to produce them, the patient has already entered the chronic phase before the serologic tests turn positive. This means, that the acute cases cannot be detected by these tests, although this would be most important, as early forms of CD infections have the best therapy outcomes”.

Because in the Study Design section the authors write that “The study focused on Dengue Fever and Fever of Unknown Origin” (L109), some references about the Fever of Unknown Origin could be added to explain the interest for the present study.

We also add here some sentences to show the difficult situation in Colombia, as only little statistics is available and cited one article. Moreover, the connection to the study title was made : “CD has two different phases: in the acute phase the patient suffers from flu-like symptoms like fever, headache and myalgias. Although the acute phase is the most promising one for a successful treatment (12, 13), CD is mostly not treated (12) due to a lack of diagnostics and is declared as fever of unknown origin. The treatment is mainly symptomatic”.

The T.cruzi genetic heterogeneity could be explained from the beginning, to link with the conclusion of the study (L529).

We also add here some information to show the difficulties in generating a good PCR.

L90-93 – the sentence is too long, a rephrase could be clarified what the authors said “… these PCRs ….. are limited in their reproducibility, and can, therefore, not be used as a reliable diagnostic tool” (ref 13, 17, 18). Someone might understand that PCRs are not recommended to be used at all. But the conclusions of the cited articles are quite different. The reference 13 (DOI: 10.1371/journal.pntd.0001689) conclusion is “These data strongly indicate that at least two PCR assays with different performances should be combined to increase the accuracy. This evaluation also highlights the benefit of extracting DNA from the blood specimen's buffy coat to increase the sensitivity of PCR analysis”.  The reference 17 (DOI: 10.1371/journal.pntd.0000931) “This study represents a first crucial step towards international validation of PCR procedures for detection of T. cruzi in human blood samples”. The reference 18 (DOI: 10.1016/j.jmoldx.2015.04.010) “This effort is a major step toward international validation of qPCR methods for the quantification of T. cruzi DNA in human blood samples…”. More in L152-153, the authors write “The TCZ- and 18S rRNA-RT-PCR were chosen due to their results in specificity (100%) (13, 17)”.

Yes, we must admit, that this paragraph was too dense. This paragraph was rephrased and we hope, it is now clearer: “Several PCRs have been developed for CD, but, as summarized in various published articles, problems occur in either sensitivity and/or specificity. Above they can vary in accuracy and performance and can have limitations in their reproducibility. (2, 3, 5, 18, 19). As the PCRs still are important and give valuable information, the recommendation is to combine at least two PCRs to come to a reliable result (2, 3, 19)”.

Materials and Methods. The inclusion /exclusion criteria are not very well synthesized.

L115 “healthy considered/asymptomatic persons”  means healthy considered or asymptomatic persons? I asked because, at least for me, it is not easy to link this information with the details from Clinical assessment (L263-273). Are these persons those 157 patients? (L272-273).  Further analysis of clinical data with tests performed in the present study would clarify the outcome of the study and clarify the authors’ discussions. The same, the 87 samples sequenced are from which persons, from a clinical point of view? (see L510, also) Therefore, clear inclusion/exclusion criteria are needed.  In the discussion section (L446-447) there is mention that “samples from the indigenous population” are tested. The explanations from L492-496 should be reflected in the Results section, also.

In persons of 12 years and above, we performed a rapid test (of course after informing and getting their permission). If this was positive, a full physical examination and a questionnaire followed. Blood was taken and analyzed further for Chagas Disease (CD), according to our protocol described. Children below 12 years went directly (after receiving the permission of their guardian) into the physical examination and blood was taken and analyzed as described before, the rapid test was then made from the taken blood.

During the anamnesis it became clear, that the indigenous have a very different body awareness than e.g. European people. They didn´t know anything about their organs, neither where they are nor what they do and above it is frowned to complain within their culture, unless it is something really severe. Therefore, most anamnesis information was very biased and difficult to use. During the physical examination we got some additional information, e.g. when asking for the reason of a scarf, of when we found e.g. an abscess on the toe, etc. Then we had also the objective information gained by the laboratory results and stool samples. We found, that many abdominal complaints were due to infections (helmints, parasites bacterial and others). Often the heart murmur could be explained e.g. by anemia, but if also CD played a role in this complaints, would have required further examinations such as ECG recording, heart echo, etc. Unfortunately this was not possible under the conditions of the study (field work). This is why missing additional examinations are a limitation of this study, and in this respect our new highly sensitive and highly specific PCR is an important improvement.

We also found, that many of those patients not stating any complaints were in fact under tough conditions, because they suffered e.g. infections with Strongyloides and Ascaris. We understand thus that a closer connection between symptoms (e.g. heart murmur) and the differential diagnoses would have been favorable, but as the information about the health status was so difficult to obtain, we rather just leave it by detailed description of the cinical state only.

Results. Serology. L283-290. So, false-positive are 9 cases, and 2 false negatives?

Yes and this was verified by sequencing.

L294- the same numbers of samples false positive in one serologic test and in two serologic tests?

We rephrase the sentence for a better understanding, It is now: “In the group of 265 non-sequenced samples, 112 were positive in at least two serologic tests”.

Please, rephrase the sentences from L314-317 for clarity.

The sentences were rephrased for a better understanding: “Out of 71 samples, being positive in at least one run and two serology results, 65 were sequenced as T. cruzi. Out of 10 samples, being positive in two serologies and the kDNA-PCR only, 8 times H. sapiens, 1 time T. rangeli and 1 time T. cruzi was sequenced. The sensitivity for kDNA-PCR within the 87 sequenced samples was found to be 89,23%, the specificity 22,73%”.

Table 1 – L324 ST: serologic test in the heading, but in the table, there are not data from the serologic tests.

Indeed they all refer to the serologic test result. We made a foodmark to further explain.

References – please write the according to the JCM requirements – “In the text, reference numbers should be placed in square brackets [ ], and placed before the punctuation…”.

Thank you for rightfully observing this. This will be corrected automatically during proof stage. We apologize, that the current reference manager used does not put square brackets.

Reviewer 3 Report

The authors developed a new qPCR assay (NDO-PCR) for the detection of T. cruzi. The newly developed qPCR method displayed superior performance over existing methods. This finding could aid timely and accurate diagnosis of Chagas Disease. The current version of the manuscript has definitely improved from the previous version and much clearer to read. Nonetheless, the manuscript may still benefit from additional minor language editing. Here are some additional comments that should be addressed.

Line 310: the detection of some of the positive samples were not reproducible in triplicate. Any explanation why this happened?

What is the LOD of the developed assay?

Author Response

The authors developed a new qPCR assay (NDO-PCR) for the detection of T. cruzi. The newly developed qPCR method displayed superior performance over existing methods. This finding could aid timely and accurate diagnosis of Chagas Disease. The current version of the manuscript has definitely improved from the previous version and much clearer to read. Nonetheless, the manuscript may still benefit from additional minor language editing. Here are some additional comments that should be addressed.

Thank you for this hint. We gave the paper for a another round to a native English speaker and in fact, this should have improved the manuscript further.

Line 310: the detection of some of the positive samples were not reproducible in triplicate. Any explanation why this happened?

This happened mainly in samples with low concentrations of DNA, being at the detection limit. However, as you may have noted, these observations are more common in the TCZ and 18s rRNA PCRs than in the kDNA and NDO PCRs.

What is the LOD of the developed assay?

We have added/changed this chapter and it is now phrased like this: “For detection limit determination, DNA from the positive control Tulahuen, taken from tissue culture, was determined in a serial dilution. The lowest dilution was defined valid when 3/3 results were positive. The detection limit (LOD) of the NDO was compared to the kDNA-PCR one magnitude superior (1:100.000 versus 1:10.000; 1,5 copies/µl versus 15 copies/µl)”.

Reviewer 4 Report

The authors claim their NDO-PCR is superior to the other three established PCR methods in a sense of specificity and sensitivity. The data are clearly shown to be fitted to their conclusion. The methodology is reasonable and discussion is also acceptable.

The paper is important by the following findings though those needs further analysis.

  1. NDO-PCR could detect the false negative due to cross-reactive T. rangeli infection which could not be detected by kDNA method.
  2. In the limited area of Colombia, T. rangeli infection is relatively dominant that may interfere the specific detection of CD by kDNA PCR.
  3. Although no data attached, the dominant strain in Colombia showed a relatively high rate of false negative by the CTZ-PCR compared with other endemic areas in Latin Americas.

This paper is highly recommended to be published in more specialized journals.

Round 2

Reviewer 1 Report

I agree with the authors that The most objective result is given when direct comparison using the same conditions is made”Unfortunately, the author's results of direct comparisons don’t agree with previously reported data of direct comparisons between TCZ and kDNA qPCR. Particularly regarding the sensitivity of TCZ qPCR. As previously stated, in two highly-cited international studies (Schijman et al. 2011 with over 300 citations and Ramirez et al. 2015 with over 100 citations) using both spiked and clinical samples for comparing qPCR methods, TCZ and kDNA qPCR were found to have similar sensitivities. Many laboratories participated in these studies to avoid biased results due to poor performance of aindividual assay in a particular settingThe current study by Kann et al. is prone to suffer from such biases. In my opinion, in this study the authors have failed to appropriately optimize TCZ qPCR. The fact that in the author's hands TCZ performs poorly does not invalidate the results obtained by NDO-PCR, which does seam to be highly-sensitive and highly-specific. Nevertheless, it undermines the novelty and potential impact of NDO-PCR since it’s improvements in sensitivity over existing highly-specific methods is probably not as large as the authors claim. Furthermore, other qPCR methods have demonstrated to be useful not only for detection, but also for quantification of parasite loads in clinical settings, while incorporating internal standards for the identification false positives (Ramirez et al. 2015). These benefits outweigh small improvements in sensitivity, again undermining the potential impact of NDO-PCR in clinical settings.

Altogether, the current study by Kann et al. lacks the novelty and impact needed to be published in the Journal of Clinical Medicine.